

# Study on the cultivation of seedlings using buds of potato (*Solanum tuberosum* L.)

Chaonan Wang[1,2,*], Chong Du[1,2,*], Zhongmin Yang[1,2], Huilin Wang[2], Leijuan Shang[2], Lili Liu[2], Zhiyi Yang[2], Shuyao Song[3] and Sikandar Amanullah[4]

[1] Xinjiang Agricultural University, Postdoctoral Station of Horticulture, Urumuqi, China
[2] Xinjiang Agricultural University, College of Horticulture, Urumuqi, China
[3] Jilin Agricultural University, College of Horticulture, Changchun, China
[4] Northeast Agricultural University, College of Horticulture and Landscape Architecture, Harbin, China
[*] These authors contributed equally to this work.

Corresponding authors
Chong Du, godv2018@163.com
Zhongmin Yang,
yangzhongmin161220@126.com

## ABSTRACT

**Background**. Potato, a vegetable crop grown worldwide, has many uses, a short growth period, a large market demand and high economic benefits. The loss of potato seediness due to traditional potato growing methods is becoming increasingly evident, and research on new ways of growing potatoes is particularly important. Bud planting technology has the advantages of more reproduction, faster growth, and simplified maintenance of crop plants under cultivation.

**Methods**. In this study, a bud planting method was adopted for the cultivation of potato seedlings. Specifically, we assessed different types of treatments for the production of high-quality buds and seedlings of potato. A total of four disease-free potato varieties (Fujin, Youjin, Zhongshu 4, and Feiwuruita) were selected, potato buds with three different lengths (3 cm, 5 cm, and 7 cm) were considered the $T_1$, $T_2$, and $T_3$ treatments, and terminal buds, middle buds, and tail buds were used as the $T_4$, $T_5$, and $T_6$ treatments. A nutrient pot experiment was performed following a randomized complete block design (RCBD) with three replicates and a natural control (CK) treatment. Cultivation was performed with the common horticultural practices of weeding and hoeing applied as needed. The photosynthetic indices, physiological indices, growth indices and quality of potato seedlings and quality of potato buds were measured at two-week intervals, and yield indices were measured when the final crop was harvested 14 weeks after planting.

**Results and Conclusions**. Cultivation of seedlings from potato buds of different lengths increased the reproduction coefficient and reduced the number of seed potatoes needed for cultivation. All morphological, physiological, and yield indices showed positive trends. A potato bud length of 7 cm was optimal for raising seedlings. Moreover, buds located at the terminal of the potato yielded seedlings with the best quality. In conclusion, we recommend that our proven bud planting technique be adopted at the commercial level, which could support good crop production with maximum yield.

## INTRODUCTION

Potato (*Solanum tuberosum* L.) is a well-known tuber crop consumed by approximately 1.3 billion people as a staple food (*Stokstad, 2019*). Potatoes are native to the Andes Mountains of South America, with an altitude of 2,000–4,000 m, where the environmental characteristics include short days, high light intensity, low temperatures and high relative humidity (*Harris, 2012*). After hundreds of years of domestication, this crop was primarily introduced into Europe in the late 1500s. It is currently being cultivated in multiple countries and regions between 65°N and 50°S (*Camire, Kubow & Donnelly, 2009*). This crop is also used as an important industrial raw material for the production of starch, ethanol and animal feed (*Birch et al., 2012*).

A potato tuber is a modified stolon that provides a rich source of nutrition due to its stored starch, protein, vitamins, and minerals (*Aien et al., 2011*; *Hancock et al., 2014*). It is an expanded stem with terminal buds and lateral buds, and the buds enter a dormant state after separating from the maternal tuber. Tuber formation begins with the inhibition of longitudinal growth at the end of the stolon, and then the terminal bud begins to expand due to cell division and cell expansion (*Cutter, 1992*). The formation of potato tubers generally depends upon favorable climatic conditions.

A number of different planting techniques have been introduced for obtaining suitable potato production. Among them, potato bud planting technology has yielded virus-free seeds with strong disease resistance characteristics at an early stage. When late blight occurs in the rainy season of July/August in the Northern Hemisphere, potato crops are being harvested or close to maturity (*Luo & Jin, 1960*). Breaking potato bud dormancy, raising seedlings in seedbeds, and transplanting seedlings into appropriate fields for rapid seed propagation could save up to 39.3% of seed potatoes (*Wu, Zhou & Min, 2009*; *Li, 2014*). When the seedling reaches a height of five cm in height, it bears a large number of fibrous roots and should be detached from the seed potato for transplanting, which could increase the reproduction coefficient of virus-free seed potato (*Xiong, Liu & Ma, 2010*).

The cultivation of potato bud seedlings is mainly achieved by seedling transplanting, which is primarily based on the cultivation of potato buds as seeds. The bud planting approach has the advantages of multiple and fast reproduction and guaranteed crop cultivation, and it has been used for the cultivation of traditional Chinese medicine crops (*Yu et al., 1999*; *Wen, Li & Chen, 2000*) and vegetable crops (*Li, 1962*). The potato bud planting method was introduced in the 1960s and has been evaluated in different provinces (Shanxi, Zhejiang, Jilin, Heilongjiang, Qinghai and other provinces) of China (*Jiang, 1959*). The growth of planted potato buds is accelerated in semihumid soil under a suitable temperature of 20–25 ° C. The length of seedlings after rapid germination can reach 3.5–10.5 cm in 8 days, and the survival rate of transplants is relatively high (*Liu, 1962*). When seedlings reach a 6–10 cm height and have 5–7 leaves, they are cut at the base and planted. The yield achieved with this method is higher than that with traditional direct seeding with buds (*Zhang & Tian, 2012*).

When the bud length of seed potato reaches 2.5 cm, crop production can increase by up to 89.4% compared to that with the conventional method of direct potato seeding (*Lv &*

*Jiang, 1992*). The cultivation of potato seedlings reduces the occurrence of potato late blight disease and reduces the probability of plant degradation (*Chen, 2014*; *Chen & Jiang, 2015*). One study showed that much older seed tuber buds produce higher yields. Bud planting reduces the number of seed potatoes needed compared with that under direct planting (*Liu et al., 1990*). The traditional method of direct bud seeding yields 200–300 kg/667 m$^2$ potatoes, but cultivation through the bud-seedling planting method requires only 40–50 kg/667 m$^2$, *i.e.,* nearly five times fewer seed potatoes (*Li, 1982*).

Vegetable seedlings are raised for vegetable cultivation (*Wen & Li, 2001*). Seedling cultivation is one of the fundamental measures for checking and improving the emergence rate and cultivating strong seedlings during vegetable production. It has numerous advantages, such as enhancing early vegetable maturity, increasing yield, and increasing economic income for farmers. At present, the protected cultivation of potato mainly involves the direct seeding of tubers, which has the crucial disadvantages of high seed consumption and a low reproduction coefficient (Fig. 1A). However, using potato buds for seedling cultivation can effectively reduce seed consumption, increase the reproduction coefficient and increase the overall efficiency of the crop. Nevertheless, there have been few reports on potato seedling cultivation through buds. The experiment reported here demonstrated the adaptability and feasibility of potato seedling cultivation using buds of different lengths in protected nutrient pots. We found that potato bud-seedling cultivation is conducive to early maturity and yield improvement.

# MATERIALS & METHODS

## Source of material and cultivation

A total of four virus-free potato varieties (Fujin, Youjin, Zhongshu 4, and Feiwuruita) were used, which were collected from the Jilin Academy of Agricultural Sciences, China. These materials yield healthy seed potatoes with excellent phenotypes, such as plant height (50~72 cm), average number of potatoes per plant (5~7), average potato weight (115~151 g) and commercial potato rate (71.3%~88%). These experimental materials were grown in a greenhouse of a practical experiment farm of Jilin Agricultural University, China. Sandy loam soil was collected and mixed with farm manure. The soil-nutrient mixture was added to pots to a depth of 30 cm. The physical and chemical properties of the soil included 1.41 g kg$^{-1}$ total nitrogen, 0.34 g kg$^{-1}$ total phosphorus, 8.9 g kg$^{-1}$ total potassium and 40.6 g kg$^{-1}$ organic matter.

## Experimental design and measurements

Seed potatoes were selected on the basis of significant morphological characteristics and a lack of visual abnormalities. Three independent trials were conducted in all cases.

(1) Raising of seedlings and cultivation of potato buds with different lengths: The potato buds were divided into three categories based on length: three cm $\pm$ 0.5 cm, five cm $\pm$ 0.5 cm, and seven cm $\pm$ 0.5 cm. Virus-free potato seeds of the same size and with strong dormancy were treated to promote germination under dark:light conditions. When the bud length reached three cm $\pm$ 0.5 cm (seed potato treatment for 8–9 d, $T_1$), five cm $\pm$ 0.5 cm (seed potato treatment for 10–11 d, $T_2$) and seven cm $\pm$ 0.5 cm

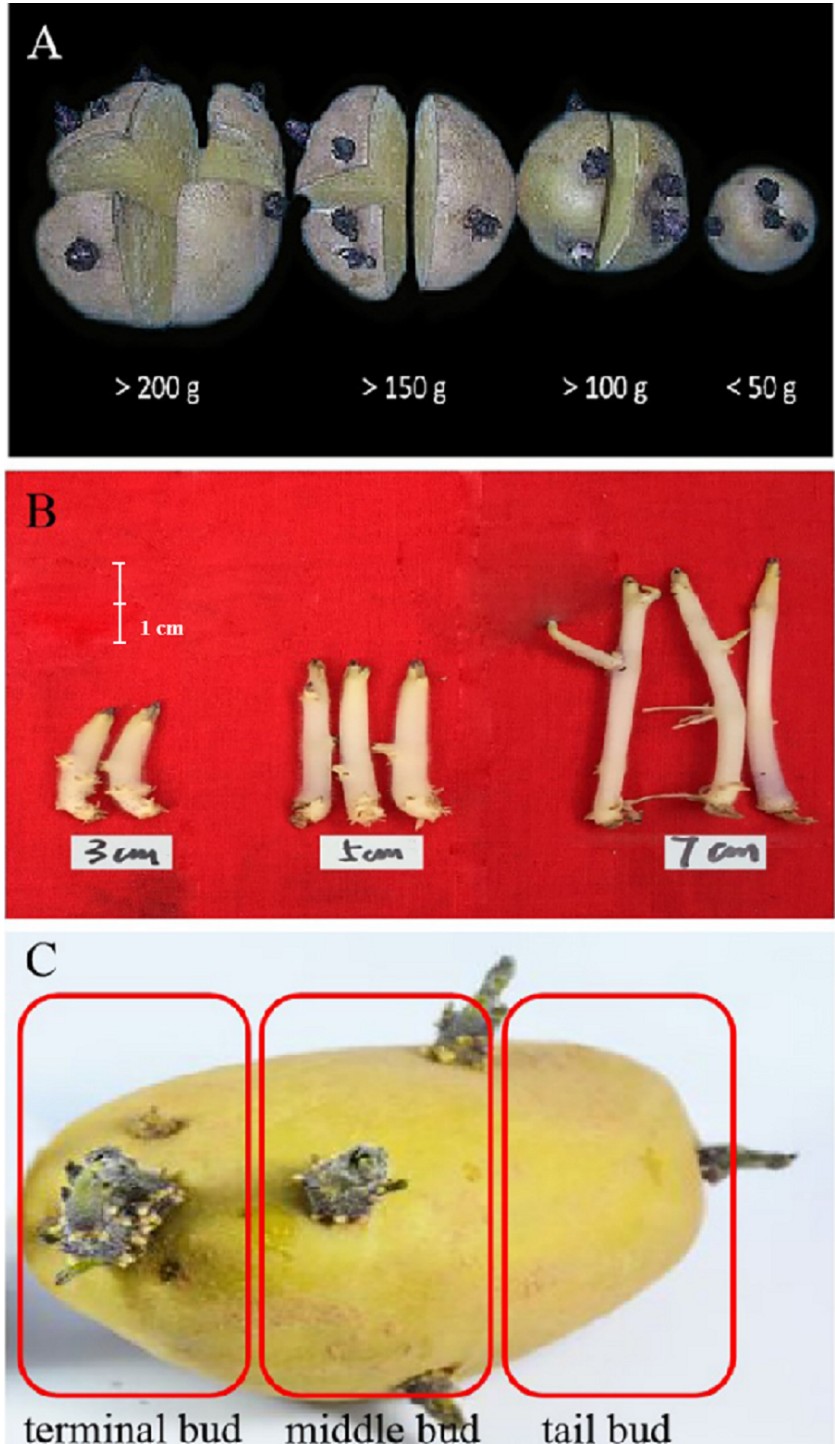

**Figure 1** **Different methods of seedling raising and cultivation of potato buds.** (A) Potato tuber treatment method for seedling cultivation with potato tuber; (B) potato buds with different lengths were used for seedling cultivation; from left to right, the potato bud lengths were three cm, five cm and seven cm, respectively. (C) Different parts of potato buds were used for seedling cultivation; from left to right, they were potato terminal bud, potato middle bud and potato tail bud, respectively.

(seed potato treatment for 12–13 d, $T_3$) (Fig. 1B), terminal buds were selected, and bud breaking treatment was performed. The buds were detached at the basal ends and sown into 10 cm × 10 cm nutrient pots under 20∼22 °C temperature, 60% air humidity, and a 14 h/8 h light-dark cycle. Seedlings were raised in a greenhouse beginning on March 25th, and 25-day-old seedlings were then planted in the greenhouse on April 20th.

(2) Raising of seedlings and cultivation of buds from various parts of potatoes: potato buds were grouped into three treatments according to their origin on the potato tuber: terminal buds ($T_4$), middle buds (middle part of the seed potato, $T_5$) and tail bud (basal bud, $T_6$). When the length of the potato bud reached five cm ± 0.5 cm (seed potato treatment for 10–11 d) (Fig. 1C), bud breaking treatment was conducted on the basal part of the potato bud, and the same described in (1) above were performed.

(3) Seed potato seedlings were cut and cultivated in nutrient pots (CK) (Fig. 1A). The seed potatoes were removed from the storage pit and placed in an incubator with light at 20 °C and 60% air humidity. When the bud length reached 1–2 cm, the bud was cultured under indoor scattered light for 2∼3 days. The seed potato was cut into bud blocks weighing 20 g and sown into a 10 cm × 10 cm nutrient pot, with one plant per bowl. The seedlings were raised in a greenhouse beginning on March 17th and transferred at 25 days of age on March 2nd. All seedlings were grown together.

A total of three experimental trials were performed in raised beds with a 1.2 m width, 5 m length, and 10 cm height, and a protective transparent film cover was applied. For double-row cultivation, the row spacing of border plants was 25 cm × 40 cm. The experiment was repeated three times following a randomized complete block design (RCBD). The environmental conditions were maintained at 17∼22 °C and 11∼13 h of light per day, and other environmental conditions, such as watering, air humidity, and fertilization, were flexibly adjusted according to growth stage.

## Sampling and measurements
### *Measurement of the photosynthesis rate and chlorophyll contents*
The photosynthesis rate of potato seedlings was determined at suitable times from 9:30 am to 11:30 am. Ten leaf samples (with each sample from the third mature leaflet from the of the plant) were carefully selected for photosynthetically active radiation (PAR) measurement. Then, the photosynthetic rate (Pn), transpiration rate (Tr), and stomatal conductance (Gs) were determined by using a LI-6400XT photosynthetic apparatus (LI-COR, Inc., Lincoln, NE, USA). The illumination intensity was 1,600 μmol photons $m^2$ $s^{-1}$, the gas flow rate was 500 mmol $s^{-1}$, the concentration of cuvette $CO_2$ was set at 400 mmol $CO_2$ $mol^{-1}$ air, and the chamber temperature was 28 °C. Then, the mixture was centrifuged with 85% (v/v) acetone solution for chlorophyll extraction, and spectrophotometric (HALO DB-20, Calamb, UK) measurements were obtained at 663 and 645 nm (*Paul & Driscoll, 1997*). Potato leaves were sampled to determine the total chlorophyll contents in plants.

### *Measurement of agronomic parameters*
Stem diameter was measured from 5 replicates of standing field plants by using an electronic Vernier caliper (instrument precision, 0.01 mm) as described previously (*Chang*

*et al., 2016*). The length and height of potato stems were measured by using a ruler with a one mm scale. A total of five plants were selected from each treatment, readings were taken three times, and the average value was calculated to determine the number of compound leaves per plant. The area of compound leaves was measured at the seedling stage and tuber expansion stage. For this purpose, five plants were randomly selected from each treatment and measured three times by using a handheld laser leaf area meter (Ci-203, CID, Inc., Vancouver, WA, USA).

Potato plants were harvested for measurements of agronomic parameters. The aboveground biomass was measured after the plants were oven-dried at 70 °C. The dry matter of detached plant organs (leaves, stems, roots, stolons and tubers) was weighed. The ratio of organ dry matter was calculated as organ (leaf, stem, root, stolon, tuber) weight/aboveground biomass × 100%; the root:shoot ratio was calculated as root dry weight/shoot dry weight × 100%; and the seedling index was calculated as (stem diameter/plant height + root dry weight/shoot dry weight) × (root dry weight + shoot dry weight).

Root activity was analyzed by the triphenyl tetrazolium chloride (TTC) method (*Wang et al., 2006*). TTC is a chemical that is reduced by dehydrogenases and mainly succinate dehydrogenase when added to a tissue. Dehydrogenase activity is regarded as an index of root activity. In brief, 0.5 g of fresh root was immersed in 10 ml of an equally mixed solution of 0.4% TTC and phosphate buffer and kept in the dark at 37 °C for 2 h. Afterward, 2 ml of $H_2SO_4$ (1 mol/L) was added to stop the reaction. The roots were dried with filter paper and then extracted with ethyl acetate. The red extractant was transferred into a volumetric flask and mixed with enough ethyl acetate to reach a total volume of 10 ml. The absorbance of the extract at 485 nm was recorded. Finally, root activity was expressed as TTC reduction intensity. Root activity was calculated as the amount of TTC reduction ($\mu$g)/fresh root weight (g) × time (h).

### Determination of potato yield and quality

After harvest, all tubers in each subplot were weighed for yield assessment using an electronic scale, and the number of tubers per plant was determined. Relative yield was calculated as the total yield of each treatment/yield of CK. The starch content of the plant sample was assayed as reported previously (*Grechi et al., 2007*). After removing the plant residue, tissue starch was extracted with 80% (v/v) ethanol. After adding 3% HCl to the residue, the spectra were determined by photometric determination at 490 nm by the phenol–sulfuric acid method. A glucose calibration curve was established to calculate tissue starch content, expressed in mg/g dry weight.

For the determination of reducing sugars, each sample was divided into three test tubes with pistons. The filtrates of centrifuged potato tuber were ground into liquid (0.2 ml), and distilled water (1.8 ml) was added to the test tubes. The method for determining the reducing sugar content was the DNS (*Miller, 1959*) method. The recorded data were used with a standard curve to calculate the reducing sugar content of ground potato tuber liquid. Crude protein was detected by Coomassie brilliant blue staining (*Gao, Wang & Luo, 2014*), and vitamin C was determined by molybdenum blue colorimetry (*Li, 2000*).

**Table 1** The photosynthetic parameter indices of different lengths of potato bud seedlings at the seedling stage.

| Treatments | | Chlorophyll content (mg/g) | Photosynthetically active radiation (PAR) $\mu mol\ m^{-2}\ s^{-1}$ | Photosynthetic rate (Pn) $\mu molCO_2$ $m^{-2}\ s^{-1}$ | Transpiration rate (Tr) m mol $m^{-2}\ s^{-1}$ | Stomatal conductance (Gs) m mol $m^{-2}\ s^{-1}$ |
|---|---|---|---|---|---|---|
| Fujin | T1 | 1.87[def] | 1012[a] | 11.25[g] | 1.9[gh] | 129[efg] |
| | T2 | 2.11[cd] | 1012[a] | 13.50[d] | 2.3[def] | 146[cd] |
| | T3 | 2.52[b] | 1012[a] | 14.07[c] | 2.7[bc] | 162[b] |
| | CK | 1.55[gh] | 1012[a] | 10.30[ij] | 1.5[ij] | 111[hi] |
| Youjin | T1 | 1.58[fgh] | 1012[a] | 11.06[gh] | 1.8[hi] | 121[gh] |
| | T2 | 1.94[de] | 1012[a] | 12.60[f] | 2.2[efg] | 138[de] |
| | T3 | 2.30[bc] | 1012[a] | 13.29[de] | 2.6[bcd] | 152[bc] |
| | CK | 1.29[h] | 1012[a] | 10.10[j] | 1.4[j] | 110[hi] |
| Zhongshu 4 | T1 | 2.30[bc] | 1012[a] | 12.78[ef] | 2.3[def] | 132[efg] |
| | T2 | 2.61[b] | 1012[a] | 14.40[c] | 2.6[bcd] | 156[bc] |
| | T3 | 3.12[a] | 1012[a] | 15.21[b] | 3.1[a] | 172[a] |
| | CK | 1.65[efg] | 1012[a] | 10.70[hi] | 1.7[hij] | 123[fg] |
| Feiwurita | T1 | 1.86[def] | 1012[a] | 13.08[def] | 2.0[fgh] | 121[gh] |
| | T2 | 2.01[cd] | 1012[a] | 15.10[b] | 2.4[cde] | 134[ef] |
| | T3 | 2.47[b] | 1012[a] | 16.03[a] | 2.9[ab] | 149[c] |
| | CK | 1.49[gh] | 1012[a] | 11.30[g] | 1.5[ij] | 109[j] |

**Notes.**
Means followed by a different letter within the column are significantly different at ($P < 0.05$) probability level according to the analysis of variance (ANOVA).

## Data analysis

The recorded data were statistically analyzed by using SPSS 20.0 statistical software (SPSS Inc., Chicago, IL, USA). One-way analysis of variance (ANOVA) with *post hoc* tests (Tukey's test) was used to detect differences among treatments. The mean values showing significant differences were compared with the Tukey test at a 5% level (*Anita, 1988*).

## RESULTS

### Effects of seedling raising on the physiological indices and growth indices of potato at the seedling stage

The chlorophyll content and net photosynthetic rate of developed seedlings with different tuber bud lengths showed the highest values for $T_3$, followed by $T_2$, and $T_1$ showed the lowest values, but they were significantly different from those in the CK treatment, as shown in Table 1. The potato seedlings were obtained by cultivating bud seedlings from different parts of the potato in nutrient pots, and the physiological indicators of each seedling stage were analyzed (Table 2). According to the chlorophyll content and net photosynthetic rate, the treatments ranked as follows: $T_4 > T_5 > T_6 >$ CK. The $T_4$ and $T_5$ treatments were significantly different from $T_6$ and CK, respectively, but no significant difference was observed between $T_6$ and CK. The net photosynthetic rate was not significantly different between nursery treatments $T_5$, $T_6$ and CK. The transpiration rate and stomatal conductance decreased in turn in the four treatments of the same variety, and the tested potato varieties showed consistency.

**Table 2  The photosynthetic parameter indices of various positions of potato bud seedlings at the seedling stage.**

| Treatments | | Chlorophyll content (mg/g) | Photosynthetically active radiation (PAR) $\mu$ mol m$^{-2}$ s$^{-1}$ | Photosynthetic rate (Pn) $\mu$molCO$_2$ m$^{-2}$ s$^{-1}$ | Transpiration rate (Tr) m mol m$^{-2}$ s$^{-1}$ | Stomatal conductance (Gs) m mol m$^{-2}$ s$^{-1}$ |
|---|---|---|---|---|---|---|
| Fujin | T4 | 2.11[b] | 1012[a] | 13.5[c] | 2.3[ab] | 146[ab] |
| | T5 | 1.94[bc] | 1012[a] | 11.5[ef] | 2.1[bc] | 130[cdef] |
| | T6 | 1.61[def] | 1012[a] | 10.6[ghi] | 1.6[de] | 117[fgh] |
| | CK | 1.55[def] | 1012[a] | 10.3[ij] | 1.5[de] | 111[h] |
| Youjin | T4 | 1.94[bc] | 1012[a] | 12.6[d] | 2.2[b] | 138[bcd] |
| | T5 | 1.54[def] | 1012[a] | 11.1[fgh] | 2.1[bc] | 127[cdefg] |
| | T6 | 1.31[gh] | 1012[a] | 10.2[ij] | 1.5[de] | 115[fgh] |
| | CK | 1.29[h] | 1012[a] | 10.1[j] | 1.4[e] | 109[h] |
| Zhongshu 4 | T4 | 2.61[a] | 1012[a] | 14.4[b] | 2.6[a] | 156[a] |
| | T5 | 2.14[b] | 1012[a] | 12.4[de] | 2.2[b] | 141[bc] |
| | T6 | 1.72[de] | 1012[a] | 10.9[ghi] | 1.8[cd] | 127[cdefg] |
| | CK | 1.65[def] | 1012[a] | 10.7[ghi] | 1.7[de] | 123[defgh] |
| Feiwurita | T4 | 2.01[b] | 1012[a] | 15.1[a] | 2.4[ab] | 134[bcde] |
| | T5 | 1.74[cd] | 1012[a] | 13.4[c] | 2.1[bc] | 120[efgh] |
| | T6 | 1.51[efg] | 1012[a] | 12.2[de] | 1.6[de] | 112[gh] |
| | CK | 1.49[fgh] | 1012[a] | 11.3[efg] | 1.5[de] | 109[h] |

**Notes.**
Means followed by a different letter within the column are significantly different at ($P < 0.05$) probability level according to the analysis of variance (ANOVA).

## Effects of seedling raising on the growth indices and quality of potato plants at the seedling stage

There were significant differences in the plant height, stem diameter, and compound leaf number of potato at the seedling stage among the treatments with different lengths of potato buds (Table 3). Seedling cultivation with seven cm potato buds was significantly longer than that with five cm and three cm buds; however, overall, the values in the three treatments were much higher than those in the CK treatment.

The $T_3$ treatment increased plant height, stem diameter and leaf number in the 'Fujin' variety by 49.63%, 60.94% and 58.81%, respectively, compared with those in CK. It increased these parameters in 'Youjin' by 21.80%, 57.83% and 43.61%, respectively, compared to those in CK; those in 'Zhongshu 4' by 49.12%, 51.09% and 40.93%, respectively; and those in 'Feiwuruita' by 37.38%, 44.7% and 35.69%, respectively. In terms of the quality of potato seedlings at the seedling stage (*i.e.,* aboveground dry weight, belowground dry weight, root activity, and seedling index), the different potato bud length treatments were ordered $T_3 > T_2 > T_1 > CK$. Most of the measured indicators in T3 were significantly different from those in CK (Table 4).

Furthermore, buds from different parts of the tuber were used for seedling cultivation in nutrient pots, and the results at the seedling stage are shown in Table 5. Within varieties, treatments ranked in terms of plant height, stem diameter and compound leaf number as $T_4 > T_5 > T_6 > CK$, with terminal buds yielding significantly higher values than observed in the CK treatment. All the tested varieties showed the same performance, with the indices of plant height, stem diameter and leaf number of 'Fujin' increasing by 32.10%, 43.94%

**Table 3 Growth indices of potato seedlings developed from different lengths of potato buds.**

| Treatment | | Plant height (cm) | Stem diameter (mm) | Number of compound leaves (piece) |
|---|---|---|---|---|
| Fujin | T1 | $9.47 \pm 0.86^{fg}$ | $7.49 \pm 0.02^{g}$ | $6.12 \pm 0.15^{g}$ |
| | T2 | $10.74 \pm 0.62^{de}$ | $9.57 \pm 0.01^{e}$ | $6.97 \pm 0.10^{e}$ |
| | T3 | $12.15 \pm 0.38^{bc}$ | $10.67 \pm 0.01^{c}$ | $8.02 \pm 0.10^{b}$ |
| | CK | $8.12 \pm 0.76^{h}$ | $6.63 \pm 0.02^{h}$ | $5.05 \pm 0.05^{j}$ |
| Youjin | T1 | $11.25 \pm 0.09^{cde}$ | $8.17 \pm 0.01^{f}$ | $6.92 \pm 0.18^{e}$ |
| | T2 | $12.14 \pm 0.15^{bc}$ | $9.43 \pm 0.01^{e}$ | $7.50 \pm 0.14^{d}$ |
| | T3 | $13.24 \pm 0.16^{a}$ | $10.18 \pm 0.01^{d}$ | $8.76 \pm 0.12^{a}$ |
| | CK | $10.87 \pm 0.26^{de}$ | $6.45 \pm 0.01^{h}$ | $6.10 \pm 0.14^{gh}$ |
| Zhongshu 4 | T1 | $9.07 \pm 0.74^{g}$ | $10.02 \pm 0.01^{d}$ | $6.04 \pm 0.12^{gh}$ |
| | T2 | $11.27 \pm 0.5^{cde}$ | $9.98 \pm 0.14^{d}$ | $6.50 \pm 0.19^{f}$ |
| | T3 | $12.78 \pm 0.20^{ab}$ | $11.12 \pm 0.04^{b}$ | $7.61 \pm 0.16^{cd}$ |
| | CK | $8.57 \pm 0.11^{gh}$ | $7.36 \pm 0.01^{g}$ | $5.40 \pm 0.14^{i}$ |
| Feiwurita | T1 | $10.28 \pm 0.15^{def}$ | $8.17 \pm 0.01^{f}$ | $6.23 \pm 0.13^{fg}$ |
| | T2 | $11.39 \pm 0.24^{cd}$ | $10.16 \pm 0.01^{d}$ | $6.80 \pm 0.13^{e}$ |
| | T3 | $13.01 \pm 0.25^{ab}$ | $11.46 \pm 0.01^{a}$ | $7.87 \pm 0.16^{bc}$ |
| | CK | $9.47 \pm 0.31^{fg}$ | $7.92 \pm 0.03^{f}$ | $5.80 \pm 0.12^{h}$ |

Notes.
Means followed by a different letter within the column are significantly different at ($P < 0.05$) probability level according to the analysis of variance (ANOVA).

**Table 4 Seedling quality of potato seedlings of different lengths measured at the seedling stage.**

| Treatments | | Shoot dry weight (g) | Root dry weight (g) | Root shoot ratio | Root activity µg/(g h) | Strong seedling index |
|---|---|---|---|---|---|---|
| Fujin | T1 | $0.89 \pm 0.09^{gh}$ | $0.21 \pm 0.02^{efgh}$ | $0.23 \pm 0.02^{bc}$ | $130.24 \pm 2.56^{e}$ | $0.34 \pm 0.03^{gh}$ |
| | T2 | $0.97 \pm 0.10^{ef}$ | $0.25 \pm 0.03^{defg}$ | $0.26 \pm 0.02^{bc}$ | $152.21 \pm 2.21^{c}$ | $0.42 \pm 0.03^{ef}$ |
| | T3 | $1.13 \pm 0.06^{cd}$ | $0.37 \pm 0.02^{ab}$ | $0.33 \pm 0.03^{a}$ | $161.07 \pm 2.78^{b}$ | $0.63 \pm 0.01^{b}$ |
| | CK | $0.84 \pm 0.08^{ghi}$ | $0.17 \pm 0.02^{h}$ | $0.20 \pm 0.01^{c}$ | $119.82 \pm 2.00^{f}$ | $0.29 \pm 0.01^{hi}$ |
| Youjin | T1 | $0.80 \pm 0.05^{hi}$ | $0.21 \pm 0.05^{efgh}$ | $0.26 \pm 0.02^{bc}$ | $127.42 \pm 2.27^{e}$ | $0.34 \pm 0.02^{gh}$ |
| | T2 | $0.89 \pm 0.07^{fgh}$ | $0.23 \pm 0.03^{efgh}$ | $0.27 \pm 0.02^{bc}$ | $139.59 \pm 1.75^{d}$ | $0.37 \pm 0.02^{fg}$ |
| | T3 | $1.16 \pm 0.07^{c}$ | $0.33 \pm 0.05^{bc}$ | $0.29 \pm 0.01^{ab}$ | $142.15 \pm 2.03^{d}$ | $0.54 \pm 0.02^{c}$ |
| | CK | $0.75 \pm 0.03^{i}$ | $0.18 \pm 0.02^{h}$ | $0.24 \pm 0.03^{bc}$ | $110.89 \pm 2.66^{g}$ | $0.27 \pm 0.02^{i}$ |
| Zhongshu 4 | T1 | $0.87 \pm 0.02^{fghi}$ | $0.20 \pm 0.01^{fgh}$ | $0.23 \pm 0.02^{bc}$ | $148.17 \pm 1.79^{c}$ | $0.37 \pm 0.02^{fg}$ |
| | T2 | $1.01 \pm 0.01^{de}$ | $0.28 \pm 0.01^{cde}$ | $0.28 \pm 0.02^{ab}$ | $163.45 \pm 2.94^{b}$ | $0.47 \pm 0.03^{de}$ |
| | T3 | $1.32 \pm 0.02^{b}$ | $0.38 \pm 0.02^{ab}$ | $0.29 \pm 0.02^{ab}$ | $170.18 \pm 2.13^{a}$ | $0.64 \pm 0.04^{b}$ |
| | CK | $0.76 \pm 0.10^{hi}$ | $0.18 \pm 0.02^{gh}$ | $0.24 \pm 0.02^{bc}$ | $129.86 \pm 1.90^{e}$ | $0.31 \pm 0.02^{hi}$ |
| Feiwurita | T1 | $0.96 \pm 0.02^{efg}$ | $0.26 \pm 0.04^{def}$ | $0.27 \pm 0.02^{bc}$ | $130.66 \pm 2.61^{e}$ | $0.42 \pm 0.02^{ef}$ |
| | T2 | $1.12 \pm 0.01^{cd}$ | $0.30 \pm 0.02^{de}$ | $0.27 \pm 0.02^{bc}$ | $148.75 \pm 4.89^{c}$ | $0.51 \pm 0.01^{cd}$ |
| | T3 | $1.56 \pm 0.02^{a}$ | $0.42 \pm 0.03^{a}$ | $0.27 \pm 0.01^{bc}$ | $153.29 \pm 1.87^{c}$ | $0.71 \pm 0.01^{a}$ |
| | CK | $0.77 \pm 0.01^{hi}$ | $0.20 \pm 0.01^{fgh}$ | $0.26 \pm 0.03^{bc}$ | $115.08 \pm 2.13^{fg}$ | $0.33 \pm 0.01^{ghi}$ |

Notes.
Means followed by a different letter within the column are significantly different at ($P < 0.05$) probability level according to the analysis of variance (ANOVA).

**Table 5  Growth indices of potato seedlings developed with various positions of potato bud.**

| Treatments | | Plant height (cm) | Stem diameter (mm) | Number of compound leaves (piece) |
|---|---|---|---|---|
| Fujin | T4 | $10.74 \pm 0.19^{def}$ | $9.57 \pm 0.02^{ab}$ | $6.87 \pm 0.09^{ab}$ |
| | T5 | $9.87 \pm 0.15^{hij}$ | $8.03 \pm 0.06^{ef}$ | $6.20 \pm 0.21^{def}$ |
| | T6 | $9.01 \pm 0.12^{jk}$ | $6.94 \pm 0.01^{hi}$ | $5.95 \pm 0.12^{ef}$ |
| | CK | $8.12 \pm 0.07^{m}$ | $6.63 \pm 0.02^{i}$ | $5.05 \pm 0.10^{g}$ |
| Youjin | T4 | $12.14 \pm 0.15^{a}$ | $9.53 \pm 0.03^{ab}$ | $7.50 \pm 0.42^{a}$ |
| | T5 | $11.54 \pm 0.36^{b}$ | $8.27 \pm 0.04^{def}$ | $6.95 \pm 0.14^{a}$ |
| | T6 | $10.98 \pm 0.44^{bcde}$ | $7.01 \pm 0.06^{ghi}$ | $6.72 \pm 0.02^{abc}$ |
| | CK | $10.87 \pm 0.26^{cdef}$ | $6.45 \pm 0.03^{i}$ | $6.10 \pm 0.14^{ef}$ |
| Zhongshu 4 | T4 | $11.27 \pm 0.50^{bcd}$ | $9.97 \pm 0.01^{a}$ | $6.50 \pm 0.21^{bcd}$ |
| | T5 | $10.29 \pm 0.19^{fgh}$ | $8.99 \pm 0.02^{bcd}$ | $6.00 \pm 0.14^{ef}$ |
| | T6 | $9.98 \pm 0.19^{hij}$ | $7.74 \pm 0.02^{efgh}$ | $5.85 \pm 0.07^{f}$ |
| | CK | $8.57 \pm 0.11^{km}$ | $7.35 \pm 0.02^{fghi}$ | $5.40 \pm 0.22^{g}$ |
| Feiwurita | T4 | $11.39 \pm 0.24^{bc}$ | $10.14 \pm 0.06^{a}$ | $6.80 \pm 0.12^{ab}$ |
| | T5 | $10.57 \pm 0.26^{efg}$ | $9.27 \pm 0.04^{abc}$ | $6.35 \pm 0.17^{cde}$ |
| | T6 | $10.02 \pm 0.12^{ghij}$ | $8.39 \pm 0.03^{cde}$ | $5.90 \pm 0.16^{f}$ |
| | CK | $9.47 \pm 0.31^{ij}$ | $7.92 \pm 0.05^{efg}$ | $5.80 \pm 0.12^{f}$ |

Notes.

Means followed by a different letter within the column are significantly different at ($P < 0.05$) probability level according to the analysis of variance (ANOVA).

and 35.29%, respectively, those of "Youjin" increasing by 11.01%, 48.44% and 22.95%, respectively, those of 'Zhongshu 4' increasing by 31.40%, 35.62% and 20.37%, respectively, and those of 'Feiwurita' increasing by 20.00%, 27.85%, and 17.24%, respectively, compared with those in the CK treatment.

The effects of bud source on potato seedling quality are shown in Table 6. The aboveground dry weight in the seedling raising treatments was higher than that in the CK treatment, and the four varieties showed consistent results. The dry weight of aboveground parts and the dry weight of belowground parts of seedlings derived from terminal buds were significantly different from those in the CK treatment. In terms of root activity and the seedling index of all tested varieties, the treatments were ordered $T_4 > T_5 > T_6 > CK$, with all varieties showing the same performance trend.

## Effect of seedling cultivation on potato plants in the greenhouse

By comparing the plant growth of potatoes grown from potato buds of different lengths during the tuber expansion stage, it was found that the three growth indicators plant height, stem diameter, and leaf area in each treatment and within the varieties were different, showing the order $T_3 > T_2 > T_1 > CK$. The treatments were significantly different, and the performance trends of the four experimental varieties were consistent.

Plant height increased by a maximum of 23.90% in the $T_3$ treatment for the 'Feiwurita' variety, and the stem diameter and leaf area increased the most in the $T_3$ treatment for 'Feiwurita', with increases of 37.32% and 16.84%, respectively (Table 7). The comparison

**Table 6  Seedling quality of potato measured at the seedling stage with various positions of potato bud.**

| Treatments | | Shoot dry weight (g) | Root dry weight (g) | Root shoot ratio | Root activity μg/ (g h) | Sound seedling index |
|---|---|---|---|---|---|---|
| Fujin | T4 | $0.97 \pm 0.01^{b}$ | $0.25 \pm 0.02^{bc}$ | $0.26 \pm 0.02^{abc}$ | $152.15 \pm 5.62^{b}$ | $0.42 \pm 0.01^{c}$ |
| | T5 | $0.94 \pm 0.01^{bce}$ | $0.20 \pm 0.01^{def}$ | $0.21 \pm 0.01^{de}$ | $139.98 \pm 6.69^{cde}$ | $0.34 \pm 0.01^{ef}$ |
| | T6 | $0.89 \pm 0.01^{cd}$ | $0.19 \pm 0.02^{ef}$ | $0.21 \pm 0.01^{e}$ | $121.54 \pm 5.92^{fgh}$ | $0.31 \pm 0.01^{ghi}$ |
| | CK | $0.84 \pm 0.01^{defg}$ | $0.17 \pm 0.01^{f}$ | $0.20 \pm 0.01^{e}$ | $119.82 \pm 2.13^{fgh}$ | $0.29 \pm 0.01^{ij}$ |
| Youjin | T4 | $0.89 \pm 0.01^{cd}$ | $0.23 \pm 0.01^{cd}$ | $0.26 \pm 0.01^{a}$ | $139.59 \pm 8.86^{cde}$ | $0.37 \pm 0.01^{d}$ |
| | T5 | $0.86 \pm 0.02^{de}$ | $0.20 \pm 0.01^{def}$ | $0.23 \pm 0.01^{cde}$ | $124.31 \pm 2.83^{fg}$ | $0.32 \pm 0.01^{fgh}$ |
| | T6 | $0.79 \pm 0.02^{efgh}$ | $0.19 \pm 0.01^{ef}$ | $0.23 \pm 0.01^{cde}$ | $115.20 \pm 4.00^{gh}$ | $0.28 \pm 0.01^{ij}$ |
| | CK | $0.75 \pm 0.02^{h}$ | $0.18 \pm 0.01^{ef}$ | $0.23 \pm 0.01^{bcde}$ | $110.89 \pm 3.13^{h}$ | $0.27 \pm 0.01^{j}$ |
| Zhongshu 4 | T4 | $1.01 \pm 0.06^{b}$ | $0.28 \pm 0.01^{ab}$ | $0.28 \pm 0.02^{a}$ | $163.45 \pm 7.00^{a}$ | $0.47 \pm 0.01^{b}$ |
| | T5 | $0.96 \pm 0.01^{bc}$ | $0.21 \pm 0.01^{de}$ | $0.22 \pm 0.01^{de}$ | $143.26 \pm 5.38^{bcd}$ | $0.36 \pm 0.02^{de}$ |
| | T6 | $0.82 \pm 0.01^{defgh}$ | $0.17 \pm 0.01^{ef}$ | $0.21 \pm 0.01^{de}$ | $135.47 \pm 6.34^{de}$ | $0.31 \pm 0.01^{ghi}$ |
| | CK | $0.76 \pm 0.01^{gh}$ | $0.18 \pm 0.01^{ef}$ | $0.24 \pm 0.01^{bcde}$ | $129.86 \pm 2.23^{ef}$ | $0.31 \pm 0.01^{ghi}$ |
| Feiwurita | T4 | $1.12 \pm 0.09^{a}$ | $0.30 \pm 0.02^{a}$ | $0.27 \pm 0.01^{abc}$ | $148.75 \pm 3.94^{bc}$ | $0.51 \pm 0.01^{a}$ |
| | T5 | $0.97 \pm 0.02^{b}$ | $0.25 \pm 0.01^{bc}$ | $0.25 \pm 0.01^{abc}$ | $137.23 \pm 5.38^{cde}$ | $0.42 \pm 0.01^{c}$ |
| | T6 | $0.84 \pm 0.01^{def}$ | $0.21 \pm 0.01^{de}$ | $0.25 \pm 0.01^{abcd}$ | $120.09 \pm 5.06^{fgh}$ | $0.35 \pm 0.01^{de}$ |
| | CK | $0.77 \pm 0.03^{fgh}$ | $0.20 \pm 0.01^{def}$ | $0.26 \pm 0.02^{abc}$ | $115.08 \pm 2.38^{gh}$ | $0.33 \pm 0.01^{efg}$ |

**Notes.**
Means followed by a different letter within the column are significantly different at ($P < 0.05$) probability level according to the analysis of variance (ANOVA).

of plant growth between potato buds collected from distinct positions during the expansion period of potato tubers is shown in Table 8. In the potato tuber expansion period, the three growth indicators plant height, stem diameter, and leaf area showed the following treatment order in each variety: $T_4 \geq T_5 > T_6 \geq CK$. The greatest plant height of 69.52 cm was observed for seedling cultivation from terminal buds of the variety 'Youjin,' the maximum stem diameter was 15.99 mm for 'Zhongshu 4', and the leaf area of 'Zhongshu 4' was the greatest at 6810 cm$^2$. The largest increases in stem diameter compared with that in the CK treatment were observed for the cultivation of 'Fujin' seedlings from terminal buds, which were 24.31%, 31.28%, and 13.55% (Table 8).

During the expansion stage of potato tubers, the dry matter distribution rate in different tissues and organs was the highest, and the order of organs from large to small was tubers, stems, leaves, roots and stolons. The distribution rate of dry matter showed the following order among the treatments with potato buds of different lengths: $T_3 > T_2 > T_1 > CK$. Within varieties, the distribution rates of tissues and organs (tubers, stems and leaves) in $T_1$ and $T_2$ were generally the same, with minor differences, but significantly higher than those in the $T_3$ and CK treatments. In the $T_1$ treatment, the percentage of tubers among varieties was approximately 33%, and the percentage of stems and leaves was approximately 64% (Fig. 2A). The percentage of 'Zhongshu No. 4' tubers was the highest (34.12%), and the proportion of stems and leaves was 63.01%. The stem and leaf accounted for the largest percentage in CK, with the highest percentage observed for 'Youjin': 80.87%.

In terms of the dry matter distribution rate in the tuber expansion stage, the treatments including plants from potato buds of different positions were ordered as followed: $T_4 > T_5$

**Table 7  Growth indices of potato seedlings with different lengths of potato buds in the potato tuber expansion stage.**

| Treatments | | Plant height (cm) | Stem diameter (cm) | Leaf area (cm$^2$) |
|---|---|---|---|---|
| Fujin | T1 | 54.09 ± 0.65$^h$ | 13.41 ± 0.19$^g$ | 6209 ± 26.88$^g$ |
| | T2 | 59.88 ± 0.97$^f$ | 15.74 ± 0.16$^d$ | 6572 ± 26.47$^d$ |
| | T3 | 64.12 ± 0.71$^d$ | 16.30 ± 0.11$^{ab}$ | 6728 ± 19.20$^c$ |
| | CK | 48.17 ± 0.82$^i$ | 12.24 ± 0.19$^i$ | 5788 ± 22.10$^j$ |
| Youjin | T1 | 62.04 ± 1.31$^e$ | 12.78 ± 0.16$^h$ | 6394 ± 11.51$^f$ |
| | T2 | 69.52 ± 0.77$^b$ | 14.96 ± 0.12$^f$ | 6799 ± 12.56$^b$ |
| | T3 | 73.08 ± 0.78$^a$ | 15.30 ± 0.11$^e$ | 6937 ± 14.98$^a$ |
| | CK | 63.27 ± 0.58$^{de}$ | 11.74 ± 0.21$^j$ | 5989 ± 13.14$^h$ |
| Zhongshu 4 | T1 | 54.13 ± 0.41$^h$ | 13.24 ± 0.18$^g$ | 6486 ± 9.20$^e$ |
| | T2 | 59.22 ± 0.87f$^g$ | 15.99 ± 0.09$^{cd}$ | 6810 ± 15.76$^b$ |
| | T3 | 65.80 ± 0.75$^c$ | 16.67 ± 0.09$^a$ | 6921 ± 33.53$^a$ |
| | CK | 47.81 ± 1.34$^i$ | 12.39 ± 0.13$^i$ | 6017 ± 9.89$^h$ |
| Feiwurita | T1 | 63.05 ± 0.69$^{de}$ | 12.78 ± 0.18$^h$ | 6456 ± 19.02$^e$ |
| | T2 | 67.29 ± 0.53$^c$ | 15.09 ± 0.14$^{ef}$ | 6710 ± 19.51$^c$ |
| | T3 | 71.84 ± 0.49$^a$ | 16.08 ± 0.12$^{bc}$ | 6926 ± 20.60$^a$ |
| | CK | 57.98 ± 0.58$^g$ | 11.71 ± 0.20$^j$ | 5928 ± 24.71$^i$ |

Notes.
Means followed by a different letter within the column are significantly different at ($P < 0.05$) probability level according to the analysis of variance (ANOVA).

**Table 8  Growth indices of potato seedlings with various positions of potato bud in potato tuber expansion stage.**

| Treatments | | Plant height (cm) | Stem diameter (cm) | Leaf area (cm$^2$) |
|---|---|---|---|---|
| Fujin | T4 | 59.88 ± 1.27$^d$ | 15.74 ± 0.33$^a$ | 6572 ± 51.23$^{cd}$ |
| | T5 | 57.54 ± 0.84$^d$ | 14.98 ± 0.29$^{ab}$ | 6420 ± 34.76$^{de}$ |
| | T6 | 49.36 ± 0.91$^f$ | 12.01 ± 0.26$^d$ | 6077 ± 75.23$^{fgh}$ |
| | CK | 48.17 ± 0.81$^f$ | 11.99 ± 0.09$^d$ | 5788 ± 41.81$^i$ |
| Youjin | T4 | 69.52 ± 0.98$^a$ | 14.96 ± 0.17$^{ab}$ | 6799 ± 92.28$^{ab}$ |
| | T5 | 68.46 ± 1.10$^a$ | 14.27 ± 0.24$^{abc}$ | 6601 ± 85.61$^{bcd}$ |
| | T6 | 64.18 ± 1.30$^c$ | 11.98 ± 0.26$^d$ | 6249 ± 11.99$^{ef}$ |
| | CK | 63.27 ± 0.93$^c$ | 11.74 ± 0.17$^d$ | 5989 ± 89.54$^h$ |
| Zhongshu 4 | T4 | 59.22 ± 1.47$^d$ | 15.99 ± 0.07$^a$ | 6810 ± 78.38$^a$ |
| | T5 | 57.14 ± 1.73$^d$ | 15.02 ± 0.21$^{ab}$ | 6507 ± 75.91$^{cd}$ |
| | T6 | 48.90 ± 1.20$^f$ | 12.94 ± 0.18$^{bcd}$ | 6200 ± 33.35$^{fg}$ |
| | CK | 47.81 ± 1.79$^f$ | 12.39 ± 0.14$^{cd}$ | 6017 ± 71.38$^{gh}$ |
| Feiwurita | T4 | 67.29 ± 1.19$^{ab}$ | 15.09 ± 0.14$^{ab}$ | 6710 ± 76.8$^{abc}$ |
| | T5 | 65.07 ± 1.03$^{bc}$ | 14.62 ± 0.20$^{ab}$ | 6459 ± 55.86$^d$ |
| | T6 | 59.82 ± 1.18$^d$ | 11.98 ± 0.08$^d$ | 6104 ± 86.37$^{fgh}$ |
| | CK | 57.98 ± 1.26$^d$ | 11.71 ± 0.13$^d$ | 5928 ± 53.55$^{hi}$ |

Notes.
Means followed by a different letter within the column are significantly different at ($P < 0.05$) probability level according to the analysis of variance (ANOVA).

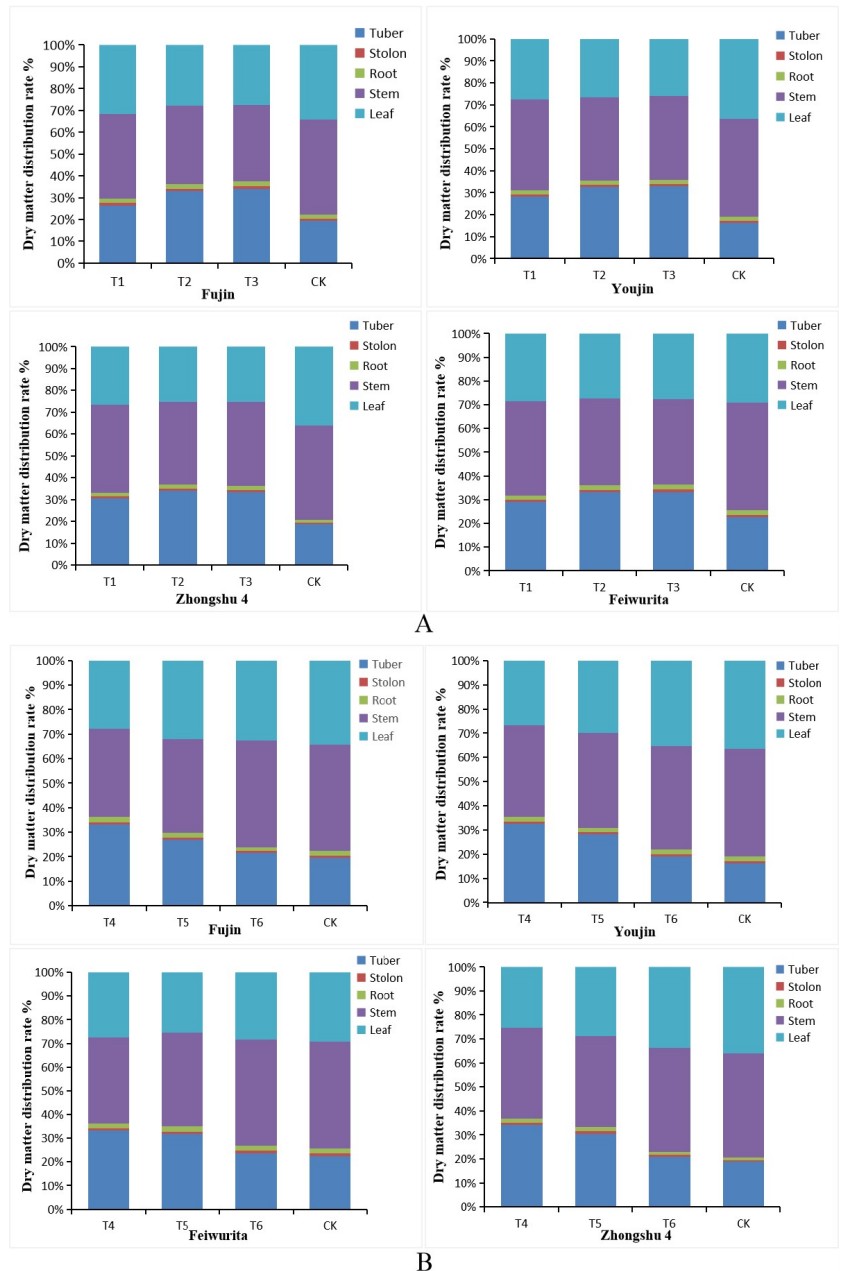

**Figure 2 Distribution rate of dry matter of different tissues and organs at the potato tuber expansion stage.** (A) Comparative effects of $T_1$, $T_2$, $T_3$, and CK; (B) Comparative effects of $T_4$, $T_5$, $T_6$, and CK.

$> T_6 >$ CK. The four tested varieties showed the same performance. The highest material distribution rate was 34.12%, and the lowest rate was 16.30%, which was observed for direct seeding of 'Youjin' buds. The stem dry matter distribution rate and leaf dry matter distribution rate of each variety were the highest in CK (Fig. 2B).

In the potato tuber expansion stage, the physiological indices (chlorophyll content and photosynthetic indices) of potato sprouts from buds of different lengths in the nutrient

**Table 9  The photosynthetic parameter index of different lengths of potato bud seedlings in the potato tuber expansion stage.**

| Treatments | | Chlorophyll content (mg/g) | Photosynthetically active radiation (PAR) $\mu$mol m$^{-2}$ s$^{-1}$ | Photosynthetic rate (Pn) $\mu$molCO$_2$ m$^{-2}$ s$^{-1}$ | Transpiration rate (Tr) m mol m$^{-2}$ s$^{-1}$ | Stomatal conductance (Gs) m mol m$^{-2}$ s$^{-1}$ |
|---|---|---|---|---|---|---|
| Fujin | T1 | 2.96 ± 0.10[fg] | 1125[a] | 16.4 ± 0.35[h] | 3.5 ± 0.19[cd] | 240 ± 7.2[de] |
| | T2 | 3.32 ± 0.07[de] | 1125[a] | 17.3 ± 0.28[g] | 3.9 ± 0.17[bc] | 251 ± 8.0[cd] |
| | T3 | 3.71 ± 0.13[bc] | 1125[a] | 18.5 ± 0.21[ef] | 4.2 ± 0.19[ab] | 259 ± 8.5[bc] |
| | CK | 2.79 ± 0.13[gh] | 1125[a] | 15.0 ± 0.24[jk] | 3.0 ± 0.13[ef] | 221 ± 6.3[f] |
| Youjin | T1 | 2.76 ± 0.10[ef] | 1125[a] | 15.6 ± 0.29[ij] | 3.3 ± 0.19[de] | 218 ± 2.8[fg] |
| | T2 | 3.12 ± 0.10[cd] | 1125[a] | 16.9 ± 0.45[gh] | 3.7 ± 0.13[cd] | 239 ± 6.6[de] |
| | T3 | 3.50 ± 0.09[i] | 1125[a] | 18.2 ± 0.43[ef] | 3.9 ± 0.13[bc] | 247 ± 10.6[cd] |
| | CK | 2.34 ± 0.14[de] | 1125[a] | 14.7 ± 0.45[k] | 2.8 ± 0.13[f] | 199 ± 9.0[h] |
| Zhongshu 4 | T1 | 3.27 ± 0.12[b] | 1125[a] | 17.4 ± 0.42[g] | 3.9 ± 0.14[bc] | 270 ± 10.6[b] |
| | T2 | 3.89 ± 0.12[a] | 1125[a] | 18.9 ± 0.29[de] | 4.3 ± 0.18[ab] | 286 ± 7.1[a] |
| | T3 | 4.26 ± 0.08[gh] | 1125[a] | 19.8 ± 0.37[bc] | 4.5 ± 0.12[a] | 290 ± 3.7[a] |
| | CK | 2.82 ± 0.06[gh] | 1125[a] | 16.2 ± 0.28[hi] | 3.6 ± 0.17[cd] | 249 ± 4.5[cd] |
| Feiwurita | T1 | 2.85 ± 0.09[gh] | 1125[a] | 19.3 ± 0.45[cd] | 3.9 ± 0.12[bc] | 227 ± 5.3[ef] |
| | T2 | 3.24 ± 0.11[e] | 1125[a] | 20.1 ± 0.28[b] | 4.2 ± 0.13[ab] | 240 ± 5.8[de] |
| | T3 | 3.60 ± 0.12[c] | 1125[a] | 20.9 ± 0.24[a] | 4.4 ± 0.18[a] | 245 ± 5.6[cd] |
| | CK | 2.61 ± 0.11[h] | 1125[a] | 18.1 ± 0.21[f] | 3.5 ± 0.14[cd] | 204 ± 4.5[gh] |

**Notes.**

Means followed by a different letter within the column are significantly different at ($P < 0.05$) probability level according to the analysis of variance (ANOVA).

pots were the highest in the T$_3$ treatment, followed by the T$_2$ > T$_1$ treatments and then CK. The maximum photosynthetic rate was 20.9 $\mu$mol CO$_2$ m$^{-2}$ s$^{-1}$ in 'Feiwurita' under the T$_1$ treatment, and the highest chlorophyll content was found in 'Zhongshu 4' under the T$_3$ treatment, which was 4.26 mg g$^{-1}$. The seedling chlorophyll content of the four tested potato varieties, ''Fujin, Youjin, Zhongshu 4 and Feiwuruita'', increased dramatically, measuring 32.97%, 49.57%, 51.06% and 37.93%, respectively, and the photosynthetic rate increased by 23.33%, 23.81%, 22.22% and 15.47%, respectively (Table 9).

During the expansion period, potato tubers showed significant differences in the chlorophyll content, photosynthetic rate, and transpiration rate depending on bud position (Table 10). Specifically, among treatments the photosynthetic rate and transpiration rate of potato during the expansion stage showed the order T$_4$ > T$_5$ > T$_6$ > CK, and T$_4$ and T$_5$ were significantly different from T$_6$ and CK, respectively. The performance trends of all varieties were similar. 'Zhong Shu No. 4' had the highest chlorophyll content of 3.89 mg g$^{-1}$, and the highest photosynthetic rate was observed in 'Feiwurita' at 20.1 $\mu$mol CO$_2$ m$^{-2}$ s$^{-1}$. The stomatal conductance of plants derived from buds at different positions was not obviously different during the expansion period. The chlorophyll content of 'Zhongshu 4' seedlings derived from terminal buds was the highest. Compared with that in CK, the photosynthetic rate of plants from these buds increased most obviously, by 37.94% and 16.67%.

**Table 10** The photosynthetic parameter index of various positions of potato bud seedlings at the potato tuber expansion stage.

| Treatments | | Chlorophyll content (mg/g) | Photosynthetically active radiation (PAR) $\mu$mol m$^{-2}$ s$^{-1}$ | Photosynthetic rate (Pn) $\mu$molCO$_2$ m$^{-2}$ s$^{-1}$ | Transpiration rate (Tr) m mol m$^{-2}$ s$^{-1}$ | Stomatal conductance (Gs) m mol m$^{-2}$ s$^{-1}$ |
|---|---|---|---|---|---|---|
| Fujin | T4 | 3.32 ± 0.14[b] | 1125[a] | 17.3 ± 0.64[bcde] | 3.9 ± 0.15[bc] | 251 ± 12.5[b] |
| | T5 | 3.18 ± 0.12[bc] | 1125[a] | 16.4 ± 0.45[efg] | 3.6 ± 0.09[cd] | 247 ± 7.7[bc] |
| | T6 | 2.84 ± 0.11[def] | 1125[a] | 15.4 ± 0.50[fgh] | 3.2 ± 0.12[e] | 229 ± 10.1[cd] |
| | CK | 2.79 ± 0.09[ef] | 1125[a] | 15.0 ± 0.57[gh] | 3.0 ± 0.13[ef] | 221 ± 8.8[de] |
| Youjin | T4 | 3.12 ± 0.19[bcd] | 1125[a] | 16.9 ± 0.78[def] | 3.7 ± 0.14[cd] | 239 ± 8.5[bc] |
| | T5 | 2.94 ± 0.17[cde] | 1125[a] | 16.2 ± 0.61[efgh] | 3.6 ± 0.14[cd] | 235 ± 10.9[bcd] |
| | T6 | 2.42 ± 0.10[g] | 1125[a] | 14.9 ± 0.56[gh] | 2.9 ± 0.21[ef] | 202 ± 6.1[f] |
| | CK | 2.34 ± 0.18[g] | 1125[a] | 14.7 ± 0.63[h] | 2.8 ± 0.09[f] | 199 ± 8.5[f] |
| Zhongshu 4 | T4 | 3.89 ± 0.16[a] | 1125[a] | 18.9 ± 0.89[abc] | 4.3 ± 0.13[a] | 286 ± 6.3[a] |
| | T5 | 3.68 ± 0.12[a] | 1125[a] | 17.8 ± 1.06[bcde] | 4.1 ± 0.20[ab] | 278 ± 2.9[a] |
| | T6 | 2.93 ± 0.09[cdef] | 1125[a] | 16.4 ± 0.65[efg] | 3.7 ± 0.10[cd] | 252 ± 8.6[b] |
| | CK | 2.82 ± 0.14[def] | 1125[a] | 16.2 ± 0.57[efgh] | 3.6 ± 0.14[cd] | 249 ± 7.1[b] |
| Feiwurita | T4 | 3.24 ± 0.14[bc] | 1125[a] | 20.1 ± 0.89[a] | 4.2 ± 0.08[ab] | 240 ± 5.6[bc] |
| | T5 | 3.08 ± 0.11[bcde] | 1125[a] | 19.4 ± 0.82[ab] | 4.1 ± 0.17[ab] | 235 ± 4.3[bcd] |
| | T6 | 2.81 ± 0.11[def] | 1125[a] | 18.3 ± 0.73[bcd] | 3.6 ± 0.20[cd] | 210 ± 6.1[ef] |
| | CK | 2.61 ± 0.14[fg] | 1125[a] | 18.1 ± 0.63[bcd] | 3.5 ± 0.09[d] | 204 ± 4.5[f] |

**Notes.**
Means followed by a different letter within the column are significantly different at ($P < 0.05$) probability level according to the analysis of variance (ANOVA).

## Effects of seedling cultivation methods on the earliness and yield of potato

The planting dates of the treatments were the same, and the harvest time and growth days in the shed were different. Within varieties, the earliest maturity was observed in $T_3$, followed by $T_2$, $T_1$ and CK. Compared with the CK, the $T_3$ treatment advanced harvest by 13 days, and the difference in harvest time between the $T_3$ and $T_2$ treatments was small, only 1–2 days (Table 11).

The yield composition of all tested potato varieties derived from potato buds of different lengths was as follows: in terms of average weight per potato and average plot yield, the difference between $T_2$ and $T_3$ was small, but it was significantly larger than that between $T_1$ and CK. In terms of average yield, the treatments ranked $T_3 > T_2 > T_1 > $ CK, the difference between $T_2$ and $T_3$ was not significant, but there was a significant difference from $T_1$ and CK. Each treatment increased the yield by more than 10%, and the $T_3$ treatment of 'Zhongshu 4' had the highest yield, reaching 3506.41 kg/667 m$^2$, an increase of 46.35% (Table 12).

The yield of cultivated potato sprouts derived from buds at various positions was measured for each variety, and it was found that compared with those in CK, the average weight per potato and average plot yield were different (Table 13). In all varieties, the indices was maximal under the $T_4$ treatment, followed by $T_5$ and $T_6$, and they were minimal in CK. The yields of 'Zhongshu 4' and 'Youjin' under the $T_4$ treatment reached 3,439.29 kg/667 m$^2$ and 3,366.53 kg/667 m$^2$, representing increases of 43.55% and 41.53%, respectively. Yield

**Table 11   Comparison of phenological periods of potato seedlings with different bud lengths.**

| Treatments | | Colonization time (month / day) | Harvest time (month / day) | Growth days in shed (day) | Relative growth days |
|---|---|---|---|---|---|
| | T1 | 4/20 | 6/18 | 58 | −7 |
| | T2 | 4/20 | 6/13 | 53 | −12 |
| | T3 | 4/20 | 6/12 | 52 | −13 |
| Fujin | T4 | 4/20 | 6/13 | 53 | −12 |
| | T5 | 4/20 | 6/17 | 57 | −8 |
| | T6 | 4/20 | 6/23 | 63 | −2 |
| | CK | 4/20 | 6/25 | 65 | 0 |
| | T1 | 4/20 | 6/20 | 60 | −6 |
| | T2 | 4/20 | 6/16 | 56 | −11 |
| | T3 | 4/20 | 6/14 | 54 | −13 |
| Youjin | T4 | 4/20 | 6/16 | 56 | −11 |
| | T5 | 4/20 | 6/20 | 60 | −7 |
| | T6 | 4/20 | 6/25 | 65 | −2 |
| | CK | 4/20 | 6/27 | 67 | 0 |
| | T1 | 4/20 | 6/16 | 56 | −8 |
| | T2 | 4/20 | 6/12 | 52 | −12 |
| | T3 | 4/20 | 6/11 | 51 | −13 |
| Zhongshu 4 | T4 | 4/20 | 6/12 | 52 | −12 |
| | T5 | 4/20 | 6/15 | 55 | −9 |
| | T6 | 4/20 | 6/21 | 61 | −3 |
| | CK | 4/20 | 6/24 | 64 | 0 |
| | T1 | 4/20 | 6/18 | 58 | −7 |
| | T2 | 4/20 | 6/13 | 53 | −12 |
| | T3 | 4/20 | 6/12 | 52 | −13 |
| Feiwurita | T4 | 4/20 | 6/18 | 58 | −12 |
| | T5 | 4/20 | 6/22 | 62 | −8 |
| | T6 | 4/20 | 6/26 | 66 | −4 |
| | CK | 4/20 | 6/30 | 70 | 0 |

**Notes.**

Means followed by a different letter within the column are significantly different at ($P < 0.05$) probability level according to the analysis of variance (ANOVA).

did not significantly increase in the $T_6$ treatment for any variety. For example, production increased by only 1.29% in 'Zhongshu 4'.

## Effects of seedling cultivation methods on potato quality

The postharvest potato quality measurements of plants from potato buds of different lengths and positions grown in nutrient pots are shown in Tables 14 and 15. The reducing sugar, starch, crude protein, and L-Vc contents did not differ between treatments in the same potato variety. However, there were significant differences in the reducing sugar and crude protein contents among the tested varieties. There was no difference in starch between 'Youjin' and 'Zhongshu 4', but there were significant differences between 'Fujin' and 'Feiwurita'. There was no difference in the content of L-Vc between 'Fujin' and

**Table 12  Effects of different lengths of potato bud seedlings on potato yield.**

| Treatments | | Number of tubers per plant (PCs.) | Average weight per potato (g) | Average plot yield (kg) | Yield per mu (kg / 667 m$^2$) | Relative yield % |
|---|---|---|---|---|---|---|
| Fujin | T1 | 4.0[ab] | 136.88[g] | 21.90[i] | 2434.72[g] | 110.27 |
| | T2 | 4.0[a] | 149.43[f] | 26.30[f] | 2923.71[d] | 132.41 |
| | T3 | 5.0[a] | 150.11[f] | 27.02[e] | 3004.13[d] | 136.05 |
| | CK | 4.0[ab] | 124.41[hi] | 19.86[k] | 2208.06[h] | 100.00 |
| Youjin | T1 | 4.0[ab] | 158.37[e] | 25.34[g] | 2816.46[e] | 116.91 |
| | T2 | 4.0[ab] | 194.10[b] | 30.28[c] | 3366.53[b] | 139.74 |
| | T3 | 4.0[ab] | 192.13[bc] | 30.74[bc] | 3417.24[ab] | 141.84 |
| | CK | 5.0[a] | 120.39[i] | 21.67[i] | 2409.17[g] | 100.00 |
| Zhongshu 4 | T1 | 4.0[ab] | 152.81[f] | 24.45[h] | 2718.16[ef] | 113.45 |
| | T2 | 4.0[b] | 203.49[a] | 30.93[b] | 3439.29[ab] | 143.55 |
| | T3 | 4.0[ab] | 187.73[c] | 31.54[a] | 3506.41[a] | 146.35 |
| | CK | 4.0[ab] | 134.69[g] | 21.55[i] | 2395.83[g] | 100.00 |
| Feiwurita | T1 | 4.0[ab] | 151.06[f] | 24.17[h] | 2687.14[f] | 120.10 |
| | T2 | 4.0[b] | 187.37[c] | 28.48[d] | 3166.53[c] | 141.53 |
| | T3 | 4.0[ab] | 180.56[d] | 28.89[d] | 3211.24[c] | 143.53 |
| | CK | 4.0[ab] | 125.81[h] | 20.13[j] | 2237.38[h] | 100.00 |

Notes.

Means followed by a different letter within the column are significantly different at ($P < 0.05$) probability level according to the analysis of variance (ANOVA).

'Youjin', but it differed significantly different from that in 'Zhongshu 4' and 'Feiwurita', respectively. In terms of the commercial potato rate, the treatments were ordered $T_3 > T_2 > T_1 > CK$ and $T_4 > T_5 > T_6 > CK$, and all tested varieties exhibited the same trend. In the experiment of seedlings from buds of different lengths and positions, the commercial potato variety that increased the most was the 'Zhongshu 4'. That in seedlings from seven cm buds increased by 5.4% compared with that in the CK, and that in the seedlings from terminal buds increased by 4.9% compared with that in the CK.

# DISCUSSION

In the current study, the bud planting method was used for the cultivation of potato seedlings. The results showed that seedling cultivation improved the physiological indices of potato plant growth, overall yield, and early maturity. Bud-seedling cultivation can significantly improve the yield of potato (Tables 12 and 13), and the result is consistent with early maturity and high yield in vegetable production (*Hai et al., 2015*). Potato bud-seedling cultivation makes full use of the main and auxiliary buds at each bud eye of the potato. This method greatly increases the reproduction coefficient of potato, which is consistent with previous findings concerning increasing the utilization rate of seed potato by repeatedly breaking off seedlings and transplanting them (*He, 1997*; *Li, Zhao & Hu, 1994*).

In this study, four potato varieties showed that seedlings from buds of different lengths can shorten the sowing period and promote early maturity and harvest compared to

**Table 13  Effects of different positions of potato bud seedlings on potato yield.**

| Treatments | | Number of tubers per plant (PCs.) | Average weight per potato (g) | Average plot yield (kg) | Yield per mu (kg / 667m²) | Relative yield % |
|---|---|---|---|---|---|---|
| Fujin | T4 | 4.0[a] | 149.43[e] | 26.30[e] | 2923.71[d] | 132.41 |
| | T5 | 5.0[a] | 137.89[f] | 24.82[f] | 2759.24[e] | 124.96 |
| | T6 | 4.0[ab] | 127.00[ghi] | 20.32[hi] | 2259.33[h] | 102.32 |
| | CK | 4.0[ab] | 124.14[i] | 19.86[i] | 2208.06[h] | 100.00 |
| Youjin | T4 | 4.0[ab] | 194.10[b] | 30.28[ab] | 3366.53[ab] | 139.74 |
| | T5 | 4.0[ab] | 179.75[c] | 28.76[c] | 3196.81[c] | 132.69 |
| | T6 | 5.0[a] | 131.28[fgh] | 23.63[f] | 2627.18[f] | 109.05 |
| | CK | 5.0[a] | 120.39[i] | 21.67[g] | 2409.17[g] | 100.00 |
| Zhongshu 4 | T4 | 4.0[b] | 203.49[a] | 30.93[a] | 3439.29[a] | 143.55 |
| | T5 | 4.0[ab] | 183.13[c] | 29.30[bc] | 3256.83[bc] | 135.94 |
| | T6 | 4.0[ab] | 136.63[f] | 21.86[g] | 2426.70[g] | 101.29 |
| | CK | 4.0[ab] | 134.69[fg] | 21.55[g] | 2395.83[g] | 100.00 |
| Feiwurita | T4 | 4.0[b] | 187.37[bc] | 28.48[cd] | 3166.53[c] | 141.53 |
| | T5 | 4.0[ab] | 170.19[d] | 27.23[de] | 3026.97[d] | 135.29 |
| | T6 | 4.0[ab] | 133.06[fgh] | 21.29[gh] | 2367.24[g] | 105.80 |
| | CK | 4.0[ab] | 125.81[ghi] | 20.13[hi] | 2237.38[h] | 100.00 |

Notes.
Means followed by a different letter within the column are significantly different at ($P < 0.05$) probability level according to the analysis of variance (ANOVA).

**Table 14  Effects of different lengths of potato bud seedlings on potato quality.**

| Treatment | | Starch (%) | Reducing sugar (%) | Crude protein (%) | VC content (mg / 100 g) | Commercial potato rate % |
|---|---|---|---|---|---|---|
| Fujin | T1 | 14.12[a] | 0.086[d] | 1.596[a] | 0.294[c] | 86.9 |
| | T2 | 14.38[a] | 0.090[d] | 1.608[a] | 0.319[c] | 88.8 |
| | T3 | 14.40[a] | 0.092[d] | 1.604[a] | 0.312[c] | 89.2 |
| | CK | 14.10[a] | 0.084[d] | 1.587[a] | 0.280[c] | 84.7 |
| Youjin | T1 | 13.55[b] | 0.063[c] | 1.395[c] | 0.288[c] | 91.7 |
| | T2 | 13.75[b] | 0.066[c] | 1.413[c] | 0.297[c] | 92.9 |
| | T3 | 13.62[b] | 0.067[c] | 1.403[c] | 0.290[c] | 93.1 |
| | CK | 13.46[b] | 0.063[c] | 1.387[c] | 0.281[c] | 90.8 |
| Zhongshu 4 | T1 | 12.44[c] | 0.426[b] | 1.518[b] | 26.601[b] | 85.1 |
| | T2 | 12.67[c] | 0.436[b] | 1.527[b] | 28.313[b] | 88.8 |
| | T3 | 12.58[c] | 0.437[b] | 1.524[b] | 27.964[b] | 89.3 |
| | CK | 12.37[cd] | 0.413[b] | 1.507[b] | 27.303[b] | 83.9 |
| Feiwurita | T1 | 12.26[d] | 1.020[a] | 1.009[d] | 123.174[a] | 87.0 |
| | T2 | 12.34 cd | 1.031[a] | 1.019[d] | 123.602[a] | 89.2 |
| | T3 | 12.37[cd] | 1.035[a] | 1.018[d] | 123.704[a] | 89.4 |
| | CK | 12.35[cd] | 1.014[a] | 1.001[d] | 122.871[a] | 86.1 |

Notes.
Means followed by a different letter within the column are significantly different at ($P < 0.05$) probability level according to the analysis of variance (ANOVA).

**Table 15  Effects of different positions of potato bud seedlings on potato quality.**

| Treatments | | Starch (%) | Reducing sugar (%) | Crude protein (%) | VC content (mg / 100 g) | Commercial potato rate % |
|---|---|---|---|---|---|---|
| Fujin | T4 | 14.38[a] | 0.090[c] | 1.608[a] | 0.319[c] | 88.8 |
| | T5 | 14.24[a] | 0.087[c] | 1.602[a] | 0.290[c] | 87.9 |
| | T6 | 14.12[a] | 0.085[c] | 1.601[a] | 0.283[c] | 86.8 |
| | CK | 14.10[a] | 0.084[c] | 1.587[a] | 0.280[c] | 84.7 |
| Youjin | T4 | 13.75[b] | 0.066[d] | 1.413[c] | 0.297[c] | 92.9 |
| | T5 | 13.60[b] | 0.065[d] | 1.409[c] | 0.290[c] | 92.4 |
| | T6 | 13.57[b] | 0.063[d] | 1.392[c] | 0.284[c] | 91.7 |
| | CK | 13.46[b] | 0.063[d] | 1.387[c] | 0.281[c] | 90.8 |
| Zhongshu 4 | T4 | 12.67[c] | 0.436[b] | 1.527[b] | 28.391[b] | 88.8 |
| | T5 | 12.56[c] | 0.427[b] | 1.520[b] | 27.942[b] | 87.2 |
| | T6 | 12.38[cd] | 0.415[b] | 1.518[b] | 27.622[b] | 85.9 |
| | CK | 12.37[cd] | 0.413[b] | 1.507[b] | 27.303[b] | 83.9 |
| Feiwurita | T4 | 12.34[d] | 1.031[a] | 1.019[d] | 126.603[a] | 89.2 |
| | T5 | 12.27[d] | 1.027[a] | 1.010[d] | 124.564[a] | 88.8 |
| | T6 | 12.29[cd] | 1.015[a] | 1.004[d] | 122.985[a] | 87.9 |
| | CK | 12.25[cd] | 1.014[a] | 1.001[d] | 122.871[a] | 86.1 |

**Notes.**

Means followed by a different letter within the column are significantly different at ($P < 0.05$) probability level according to the analysis of variance (ANOVA).

those observed with potato tuber seeding (Tables 12 and 13). Among the treatments, $T_3$ and $T_2$ exhibited the best performance, mainly because the bud seedling cultivation process requires a brief period of time to grow seedlings, and the plants are grown in advance (*Fang, 2019*). Potato plants grow faster at the early stage, and the underground root system develops, resulting in a larger leaf area, robust photosynthesis, more dry matter accumulation, and potential plant growth (*Zhou, Wu & Li, 2014*). Potato plants can enter the tuber swelling stage before this time, and then the tuber grows and develops in the ground for a long time, resulting in high yield. Furthermore, the temperature is lower at earlier stages of tuber development, which is suitable for tuber growth. A large temperature difference between day and night is conducive to tuber expansion and the movement of stem and leaf assimilates to tubers (*Zhang, 2012*). Therefore, the yield of potato bud seedling cultivation was significantly higher than that in the tuber seedling treatment.

Under the same management conditions, plant height and leaf number in the potato seedling stage (Tables 3 and 5) were consistent with the standards of a plant height of 5–10 cm and 5–8 leaves in the seedling transplanting method (*Wang, 2009*). There was no significant difference in yield performance between potatoes from five cm bud seedlings and seven cm bud seedlings, but there was a significant difference compared with three cm bud seedlings, which was positively correlated with the growth and vigor of potato buds (Table 12). At the seedling stage, some buds from the terminal of the potato grew robustly, showing maximum values for plant height, stem diameter and compound leaf number (Table 3). This finding was consistent with the principle of apical dominance in potato, where the yield of terminal buds is higher than that of buds from other positions (*Wen*

& Wang, 1993; Li, 2003). Young leaves of potato plants from terminal buds appear earlier and grow faster, the seedlings grow vigorously, and the yield is significantly increased, which is related to the early growth and development of the plants. The main advantage of the terminal buds is obvious (Wen & Wang, 1993). The leaf number of potatoes generated from middle buds, tail buds and the CK treatment decreases consecutively, the growth potential decreases, the leaf development of plants from young buds is delayed, and the growth rate slows down, consistent with potato tuber bud-breaking propagation (Xiao & Guo, 2007).

At the same time, it was proven that young leaves of potato terminal buds started early and grew faster, seedlings grew vigorously, apical dominance of terminal buds was obvious, and the yield increased significantly. It seems to be related to the growth and development of plants at the early stage, while the number of leaves in the potato middle bud, potato tail bud and control treatments decreased one by one, the growth potential decreased in turn, the leaf spreading time of the young bud became later, and the growth rate slowed down, which is consistent with the bud breaking propagation of potato tuber (Xiao & Guo, 2007).

In the various experimental treatments of potato bud seedling cultivation, the management methods, fertilizer application and water management during the growth period were consistent with those applied in greenhouse cultivation. Therefore, there was no difference in quality between the treatments of various varieties. Nevertheless, there were no differences among the varieties, consistent with potato quality and varietal characteristics. In the current experiment, five cm potato buds were selected for seedling cultivation, with cultivation of seedlings from terminal buds yielding the best result. In short, three potato bud lengths, namely, 3 cm, 5 cm, and 7 cm, were selected according to performance in the potato bud length test. There may be other suitable approaches for potato cultivation involving other bud lengths and positions, which needs to be further investigated. In addition, assessment of other gradients of potato sprout lengths for seedling treatment is required.

## CONCLUSIONS

In this study, the potato bud planting method was used for the cultivation of seedlings. Potato buds with three different lengths (3 cm, 5 cm, and 7 cm) were considered the $T_1$, $T_2$, and $T_3$ treatments, and terminal buds, middle buds, and tail buds were used as the $T_4$, $T_5$, and $T_6$ treatments. The plant morphological and physiological growth indices were significantly different among the treatments, and the seven cm potato bud length treatment ($T_3$) was proven to be the best treatment for raising seedlings under simulated field conditions. Among all tested treatments, $T_4$ showed excellent growth, followed by $T_5$, $T_6$, and CK, during the seedling stage and all growth stages. There were also significant differences among the treatments of the tested varieties during the determination of yield and commercial potato production rate. Regarding the bud position treatments, terminal buds were shown to be the best treatment because the relative yield and commercial potato yield rates were the highest. In conclusion, we demonstrated that our evaluated bud planting technique should be adopted at the commercial level, which could help achieve enhanced crop production with maximum yield.

## ACKNOWLEDGEMENTS

We thank Chaonan Wang, Chong Du and Shuyao Song for conceiving and designing the experiments. We thank Zhongmin Yang, Huilin Wang, Leijuan Shang, Lili Liu, and Zhiyi Yang for analyzing the data. We thank Sikandar Amanullah for writing, reviewing and editing this manuscript.

### Funding

This work was supported by the Natural Science Foundation of Xinjiang Uygur Autonomous Region (No. 2022D01B27), the Tianchi PhD Program in the Xinjiang Uygur Autonomous Region (No. 390000017) and the crosswise tasks project (No. 2521HXKT1). The funders had no role in study design, data collection and analysis, decision to publish, or preparation of the manuscript.

### Grant Disclosures

The following grant information was disclosed by the authors:
Natural Science Foundation of Xinjiang Uygur Autonomous Region: 2022D01B27.
Tianchi PhD Program in the Xinjiang Uygur Autonomous Region: 390000017.
crosswise tasks project: 2521HXKT1.

### Competing Interests

The authors declare there are no competing interests.

### Author Contributions

- Chaonan Wang conceived and designed the experiments, performed the experiments, prepared figures and/or tables, authored or reviewed drafts of the article, and approved the final draft.
- Chong Du conceived and designed the experiments, prepared figures and/or tables, authored or reviewed drafts of the article, and approved the final draft.
- Zhongmin Yang conceived and designed the experiments, authored or reviewed drafts of the article, and approved the final draft.
- Huilin Wang analyzed the data, authored or reviewed drafts of the article, and approved the final draft.
- Leijuan Shang analyzed the data, prepared figures and/or tables, and approved the final draft.
- Lili Liu analyzed the data, prepared figures and/or tables, and approved the final draft.
- Zhiyi Yang analyzed the data, authored or reviewed drafts of the article, and approved the final draft.
- Shuyao Song performed the experiments, authored or reviewed drafts of the article, and approved the final draft.
- Sikandar Amanullah analyzed the data, authored or reviewed drafts of the article, edited and embellished the language of the manuscript, and approved the final draft.

## Data Availability

The original measurement values are available in the Supplemental Files.

## Supplemental Information

Supplemental information for this article can be found online at http://dx.doi.org/10.7717/peerj.13804#supplemental-information.

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
