# Peer review of "Study on the cultivation of seedlings using buds of potato (Solanum tuberosum L.)"

_PeerJ, doi:10.7717/peerj.13804_

## Round 0.1 · original submission · Major Revisions

Please carefully evaluate the reviewers' comments and modify the text according to their indications. In agreement with the authors, a revision of the English is suggested.

·

Basic reporting

Reviewer comments:
The experimental work entitled “Study on the cultivation of seedlings using buds of potato (Solanum tuberosum L.)” is an instant study engaged with emphasis on analysis of morpho-physiological attributes and crop production-related technique in potato. It seems that the authors carried out relatively good experimental methods and obtained satisfactory results for the cultivation of seedlings using buds of potato (Solanum tuberosum L.). I think that such type of study is really needed for the discovery of crop production-related innovative techniques. This manuscript seems to fall within the universal scope of PeerJ Journal. However, a minor revision is necessary before the acceptance of this manuscript.
My reviewed comments to this manuscript (ms) are listed below, please go through the entire ms and correct the followings:

1. Treatments abbreviations should be written in subscript style.
e.g. Line 34. Please use T1, T2, T3, T4, T5, and T6.

2. Please add the citation for each method that you referred to and check all citations.
e.g. Line 241-244. Insert the citation for software SPSS 20.0 used for statistical analysis in the materials and methods section. Also, add its reference in the reference section.

3. Please use the appropriate units for measured traits and the format of each unit should be uniform in the entire ms.
Please amend! e.g.
Line 98 ‘200-300 kg/mu’ and Line 340 ‘3506.41 kg/667m2’ (1 mu = ~667 m2).
Line 134 ‘8-9d’ and line 138 ‘25 days’. Please use either full or abbreviated form (8-9 d, 25 d would be good).
Line 150 ‘2 March’. Please use the Month/Date format (March 2nd).
Line 152 ‘1.2m’ and ‘5m’. Please add space between values and units (1.2 m, 5 m).
Line 168 and 194 ‘°C’. Please pay attention to the font formats.
Line 310 and 323 ‘μMol CO2 m-2 s-1’ and ‘μmol CO2 m-2 s-1’. Please check the capital or lower-case letters and subscript or superscript (μmol CO2 m-2 s-1).
I recommend adjusting all the unit formats in the ms into a uniform one ‘xa y-b’ (g/kg to g kg-1, m2/s to m2 s-1, mol/L to mol L-1, etc.).

4. Description should be consistent throughout the main text as well as legends.
Please correct! e.g.
You described the last bud as the bottom bud in the legend of Figure 1. However, you described that bud as the tail (basal) bud (Line 402) in the main text. Please amend!
In addition, please add a scale bar in Figure 1A.
Line 45-46 Results and Conclusions section. The length of ‘7 cm’ potato bud was proven as the best treatment among the different treatments used for raising of seedlings. However, Line 430-432 The obtained results depicted that morphological and physiological growth indexes of plants were significantly different among the treatments, and ‘5cm’ potato bud length treatment (T2) was proven as the best treatment for raising of seedlings and in line with actual requirement. Inconsistent!
I believe it’s 7 cm should be the best treatment. Please double-check the results and avoid the wrong description.

5. Please check the entire reference section and list all the references in a uniform format as required by PeerJ Journal.

6. Grammar and writing issues, please revise these issues in the entire ms. Here, I only listed some that I found while reading.
Please check and amend! e.g.
Line 58 hundreds years of its domestication/hundreds of years of domestication???
Line 67 is generally depends upon/generally depends upon (check Line 105)
Line 69 has been/have been
Line 87 3.5-10.5 cm growth? Length?
Line 109 A very little is reported/little has been reported
Line 120 The cultivation of these experimental materials was grown/These experimental materials were grown
Line 143 described in above part 1/described above in part 1
Line 306-309 The sentence is too long and confusing, please describe it more concisely and clearly!
Line 322 It’s not a sentence. ‘Among them, the chlorophyll content of ‘Zhongshu 4’ with the highest rate of 3.89 mg/g’.
Line 334-340 This paragraph needs to be revised.
Line 341-348 This paragraph needs to be rewritten, including punctuation use (Line 341-342), grammar issues (Line 343-345) and ways of description (Line 346-348).
Your description: the T6 treatment of each variety did not increase the yield significantly, ‘Zhongshu 4’ only increased production by 1.29%.
Revised: The yield was not significantly increased in T6 treatment of each variety. For example, production was only increased by 1.29% in ‘Zhongshu 4’.
Please amend all the similar issues throughout the ms!

Experimental design

no comments

Validity of the findings

no comments

Additional comments

no comments

·

Basic reporting

The research paper has been clear; professional English has been used throughout the manuscript. Literature references are well-cited across the manuscript and in context.
Figures and tables are confirmed to the manuscript objectives.

Experimental design

The research question of the manuscript is well explained throughout the manuscript. The research question has been performed with ethical and technical standards. The method has been detailed and described sufficiently.

Validity of the findings

The manuscript shows impactful and meaningful findings in potatoes cultivation, and the agronomic approach adopted by the authors is in line with agronomic standards.

·

Basic reporting

The manuscript "Study on cultivation of seedling using buds of potato (Solanum tuberosum L.)" compares morphological, physiological and yield performance of potato seedling obtained from bud and cut tubers, and cultivated in beds.

The manuscript needs an extensive English editing.

Experimental design

In general, the experimental design is well planned, however, from the point of view of potato production, I have objections with respect to the authors' consideration of "control" and "commercial level".
1-The framework in which the authors introduce its study is the potato bud planting technology, however, around the world, the common planting method is the use of seed potatoes (whole minitubers or tubers cut into pieces) planted in the ground. Unfortunately, I have not access to read the vast bibliography that the authors presents respecto to the bud planting technology, because of these works are in Chinese (Luo and Jin, 1960; Wu et al., 2009; Li, 2014; Xiong et al., 2010; Yu et al., 1999; Wen et al., 2000; Li, 1962; Li, 1959).
In this sense, I think that the more appropriate control would have been the use of a seed potato and not the use of seedling obtained from cut tubers.

2-Respec to the author's conclusion that the assayed bud planting technique should be adopted at commercial level, this must be taken with caution. The experimental trials consisted of raised beds of 1.2m width, 5m length, 10cm height covered with a protective transparent film, quite different conditions than the ones used at commercial level.

Validity of the findings

The work is hard to read from the results section onward, since it makes a systematic error by confusing the information presented in the text with respect to that shown in the tables. For example, in Table 1, T3 treatment presents the highest values, while in the text, the authors indicate that "…showed highest values at T1, followed by T2, and T3 showed the lowest values…" (L227-229). The authors must make an extensive revision and correction of all presented results according to the data exposed in Tables.

Additional comments

Particular comments
-L59. Please verify the year, the potato was not introduced into Europe in 1970.
-L72. Please, give the hemisphere.
-L119. What are excellent phenotypes? Give more precisions.
-L152-158. Please, give separation between plants.
-L180. How do the authors determine the tuber expansion stage?
-L202. It is not clear what is a sub-plot. Please clarify.
-L252-253. There are characters in which T1 was not different from CK. Please verify.
-L277. In Feiwurita, T2 is higher than T1 for plant height. Please verify.
-L278. Replace thickness by diameter.
-L284. Please verify, the var. Eugene was not evaluated in this work. The same for L345.
-L359-360. Verify this assertion. There are treatments that showed the same response.
-L283. Please, verify this information: IN GENERAL, T4=T5>T6=CK.

---

## Round 0.2 · Minor Revisions

The authors did a great job following carefully the reviewers' suggestions. In accordance with the indications of the Rev#1, I suggest a "minor revision", that takes into account a check of the English.

·

Basic reporting

The authors modified the issues that I mentioned in my last comments, including unit and date format, full and abbreviate form. They also corrected grammar/writing issues and rewrote the paragraphs according to my comments.
The ms has been well revised but still needs further English editing!

Experimental design

no comment

Validity of the findings

no comment

·

Basic reporting

The authors have substantially changed the article according to the inputs presented by the reviewers.

Experimental design

The experimental design is coherent with the Aims and Scopes of the journal.

Validity of the findings

The scientific novelty is present and lies in the fact that bud planting could be very useful in developing future cultivation systems for potato plants.

Additional comments

No additional comments.

---

## Round 0.3 · accepted · Accept

The authors did a great job, the paper is ready to be accepted.